# Targeting Mitochondria in Diabetes

**DOI:** 10.3390/ijms22126642

**Published:** 2021-06-21

**Authors:** Nina Krako Jakovljevic, Kasja Pavlovic, Aleksandra Jotic, Katarina Lalic, Milica Stoiljkovic, Ljiljana Lukic, Tanja Milicic, Marija Macesic, Jelena Stanarcic Gajovic, Nebojsa M. Lalic

**Affiliations:** Clinic for Endocrinology, Diabetes and Metabolic Diseases, University Clinical Center of Serbia, Faculty of Medicine, University of Belgrade, Dr Subotica 13, 11000 Belgrade, Serbia; nina.krako@med.bg.ac.rs (N.K.J.); kasja.pavlovic@med.bg.ac.rs (K.P.); aleksandra.z.jotic@gmail.com (A.J.); katarina.s.lalic@gmail.com (K.L.); mmstoiljkovic@yahoo.com (M.S.); ljikson17@gmail.com (L.L.); icataca@gmail.com (T.M.); macesicmarija@gmail.com (M.M.); stanarcicjelena@gmail.com (J.S.G.)

**Keywords:** type 2 diabetes, insulin resistance, mitochondria, respiration, respiratory capacity, liver, skeletal muscle, blood cells, exercise, diabetes therapy

## Abstract

Type 2 diabetes (T2D), one of the most prevalent noncommunicable diseases, is often preceded by insulin resistance (IR), which underlies the inability of tissues to respond to insulin and leads to disturbed metabolic homeostasis. Mitochondria, as a central player in the cellular energy metabolism, are involved in the mechanisms of IR and T2D. Mitochondrial function is affected by insulin resistance in different tissues, among which skeletal muscle and liver have the highest impact on whole-body glucose homeostasis. This review focuses on human studies that assess mitochondrial function in liver, muscle and blood cells in the context of T2D. Furthermore, different interventions targeting mitochondria in IR and T2D are listed, with a selection of studies using respirometry as a measure of mitochondrial function, for better data comparison. Altogether, mitochondrial respiratory capacity appears to be a metabolic indicator since it decreases as the disease progresses but increases after lifestyle (exercise) and pharmacological interventions, together with the improvement in metabolic health. Finally, novel therapeutics developed to target mitochondria have potential for a more integrative therapeutic approach, treating both causative and secondary defects of diabetes.

## 1. Introduction

Diabetes is a chronic metabolic disease caused by insufficient insulin production and secretion and, in case of type 2 diabetes (T2D), the inability of tissues to adequately respond to insulin. These changes result in a high concentration of glucose in the blood, which in time leads to different complications. Diabetes is one of the most prevalent noncommunicable diseases today—according to the 2019 International Diabetes Federation Diabetes Atlas, there are 463 million people in the world suffering from diabetes [1].

Pathogenesis of T2D can be explained by the triumvirate hypothesis, according to which the dysfunction of three key organs—decreased insulin secretion by the pancreas, increased hepatic glucose production and decreased muscle glucose uptake—cumulatively leads to hyperglycemia [2]. This hypothesis was later expanded to include other tissues that contribute to glucose homeostasis, such as adipose tissue, kidney, gut, α-cells of the pancreas and the nervous system, which together with the previous three complete the so-called ominous octet of diabetes [3]. At the level of tissues and cells, the basic pathophysiological mechanism underlying the disturbance of normal metabolic function is insulin resistance (IR)—an inability of tissues to respond to the insulin signal. At first, the pancreas compensates by overproducing insulin, but at some point in the disease development, insulin secretion starts to decrease as well. IR primarily affects the liver, skeletal muscle and adipose tissue, which results in inadequate substrate handling by these tissues. Instead of storage and oxidation of glucose and lipids in the fed state, these tissues act as if the organism is fasting—by breaking down reserves and exporting glucose and fatty acids into the bloodstream. This substrate overload leads to hyperglycemia, ectopic fat accumulation and consequently to further worsening of IR.

Mitochondria play a central role in the cellular energy metabolism—they are responsible for oxidative phosphorylation (OXPHOS) and β-oxidation of fatty acids, which makes them essential for glucose and lipid metabolism and adenosine triphosphate (ATP) production. Mitochondrial dysfunction has long been associated with diabetes, as a potential cause of both IR and β-cell dysfunction [4,5], and more recently in the context of secondary diabetes complications [6]. There are numerous hypotheses explaining the association of different aspects of abnormal mitochondrial function to diabetes—from decreased mitochondrial content and disturbances in mitochondrial biogenesis [7], to impaired mitochondrial function leading to intracellular accumulation of lipid products and increased production of reactive oxygen species (ROS) [8], that all further impair insulin sensitivity and energy metabolism (reviewed in detail by Patti and Corvera) [9]. Having all this in mind, important questions remain unanswered to this day—we still do not know if mitochondrial dysfunction is the cause or consequence of diabetes and when in the course of the disease it appears—whether it precedes IR and hyperglycemia or it comes later.

Certain mitochondrial disorders—inherited diseases caused by mutations in the mitochondrial genome—that cause a diabetes phenotype, for example MELAS (mitochondrial encephalopathy, lactic acidosis and stroke-like episodes) and MIDD (maternally inherited diabetes and deafness), make an interesting case for exploring the relationship between mitochondrial dysfunction and diabetes. Clinical features that raise suspicion for mitochondrial diabetes include multi-organ involvement, elevated serum lactate levels, a more rapid progression to insulin therapy and the earlier onset of diabetes-related complications compared to individuals with T2D. The inability of dysfunctional mitochondria to produce sufficient ATP results in multi-organ defects and affects predominantly organs with high energy requirements such as the central nervous system, muscle, retina, kidney and pancreas [10].

Mitochondria in diabetes are studied and described by many different methods and parameters in different studies. Some methods measure mitochondrial content (number, volume, activity of marker enzymes or mtDNA copy number), morphology or parameters related to mitochondrial biogenesis (expression of mRNA or proteins involved in these processes); others measure mitochondrial function, which can also be done in many ways: protein abundance of respiratory chain complex subunits, in vitro activity of different mitochondrial enzymes, ex vivo oxygen consumption (respirometry), ATP synthesis or in vivo measurement of ATP or phosphocreatine resynthesis rates [11,12]. As the results obtained by different methods are difficult to compare and draw a clear conclusion from, we focus mainly on the studies that measure ex vivo respirometry with samples from human subjects. We apply a novel, harmonized terminology concerning mitochondrial respiratory states and rates, where applicable, that was recently published by a consortium of researchers [13]. When discussing previous work where authors used the older terminology, we replaced it with the new one (state 3—OXPHOS capacity, state 4—LEAK respiration, state u/uncoupled—ET capacity) for the purpose of simplicity and harmonization.

Since mitochondrial dysfunction is present in different tissues and contributes to diabetes pathogenesis and complications in a plethora of ways, a therapeutic intervention that targets mitochondria could present a more integrative therapeutic approach, treating different causative and secondary defects of diabetes at once. This review aims to give an overview of the latest studies done in human subjects with T2D that investigate mitochondria in skeletal muscle, liver and blood cells as a potential target for diabetes therapy. Even though this topic has been extensively studied and reviewed [14,15,16] we wanted to focus on in vivo and ex vivo human studies using respirometry to asses mitochondrial function directly, as these could be compared and analyzed easily, without the problem of multiple different methodologies used. Additionally, we wanted to summarize how some of the commonly used interventions for diabetes and related diseases affect mitochondrial function in diabetes and to comment on the potential of novel drugs targeting mitochondria in diabetes therapy. Among all insulin-sensitive tissues, we decided to focus on the role of mitochondria in skeletal muscle and liver (as major sites for glucose disposal and production), since insulin resistance and mitochondrial dysfunction in these tissues have the biggest impact on the disruption of whole-body energy homeostasis. Along with these tissues commonly studied in diabetes, we include in this review studies done on blood cells, as we want to point out the potential of respirometric analysis of these samples in drug screening and surrogate marker validation. We also review studies of lifestyle (exercise) and pharmacological interventions in humans with T2D that assess mitochondrial respiration and summarize new perspectives in diabetes therapy with a focus on mitochondria.

## 2. Role of Mitochondria in Diabetes

### 2.1. Skeletal Muscle

Mitochondrial function has been widely studied in diabetes, especially in skeletal muscle. As a highly metabolically active tissue, constituting the highest percentage of total body mass, skeletal muscle is, together with liver, a key player in glucose homeostasis regulation, and it is highly impacted by metabolic perturbations such as IR and metabolic syndrome. It is also the main tissue responsible for whole-body metabolic benefits of physical activity and exercise intervention (Section 3.1).

When studying mitochondrial function in human muscle, there are multiple ways to do so. Mitochondrial function can be studied in vivo, using magnetic resonance spectroscopy [17], or ex vivo, using high-resolution respirometry for measuring mitochondrial function in tissue biopsies or isolated mitochondria [18]. Skeletal muscle biopsies are, compared to some other tissues, easy to obtain and less invasive for the study subjects. High-resolution respirometry is based on measuring the concentration of oxygen in a closed chamber, where oxygen flux is used as a measure of mitochondrial respiratory function.

The first studies that explored muscle mitochondrial function in diabetes measured the activity of oxidative and glycolytic enzymes in skeletal muscle tissue. They came to the conclusion that in diabetes, glycolysis was favored and oxidative phosphorylation was downregulated, perhaps as an adaptive mechanism for increased substrate availability [19,20]. Disruption of normal mitochondrial dynamics is one of the most frequently proposed mechanisms behind mitochondrial dysfunction in diabetes—numerous studies show decreased expression of the transcription factor peroxisome proliferator-activated receptor gamma coactivator 1α (PGC-1α) but also decreased expression of PGC-1α and nuclear respiratory factor-1 (NRF-1) responsive genes that encode oxidative enzymes [21,22]. Later studies that measure mitochondrial function directly, in vivo or ex vivo, show a decrease of mitochondrial respiration [23,24], but many more show that when normalized to mitochondrial content (citrate synthase activity or mitochondrial DNA copy number) respiration does not differ between subjects with diabetes and healthy controls and that mitochondrial dysfunction in diabetic muscle is a consequence of lower mitochondrial content and not lower intrinsic mitochondrial function [25,26].

Other aspects of respiratory function have also been a subject of some studies. Rabol et al. [27] show a decreased respiratory control ratio (RCR), indicating lower coupling efficiency, but normal maximal respiration, in patients with poor glycemic control, which was improved by intensive insulin treatment. Larsen et al. [26] show an unchanged mitochondrial respiratory capacity, when normalized to mitochondrial content, but a higher sensitivity to respiratory substrates for complex I and II of the electron transport chain. Interestingly, this higher sensitivity was not present for fatty acids as respiratory substrates, and the authors hypothesize a possible adaptation to low glucose in the diabetic, insulin-resistant muscle, which would not appear when using fatty acids because of the high availability of this substrate in the diabetic muscle, also prone to intramuscular fat accumulation.

Taken together, it could be concluded that there is an undeniable correlation between T2D and IR on the one side and mitochondrial dysfunction on the other. It is not yet clear if this is due to a decrease in intrinsic mitochondrial function or to a decrease in mitochondrial content, since the results of numerous studies exploring this subject are not coherent. Different underlying mechanisms are proposed as a cause of this, the most common being a disruption of lipid metabolism in the mitochondria and consequential intramuscular fat accumulation and build-up of lipid products such as diacylglycerol and ceramides, which have been shown to inhibit insulin signaling by activating novel protein kinases C (PKCs) (like PKCθ) [28]. Another hypothesis is that impaired oxygen supply to the skeletal muscle, and not intrinsic impairment of mitochondrial oxidative function, is a cause of suboptimal muscle mitochondrial function [29]. As in other tissues, increased ROS production is present in the diabetic muscle, potentially as a consequence of abnormal mitochondrial function, but further propagating a disturbance in function as well [30].

### 2.2. Liver

The liver plays a major role in glucose homeostasis regulation, releasing glucose during fasting and storing it in the form of glycogen after meals. In the insulin resistant liver, insulin fails to suppress endogenous glucose production when glucose is abundant, and instead of storing, keeps releasing it into the circulation. A strong correlation has also been found between liver fat accumulation and fasting serum insulin, which might be in part due to a decrease in hepatic insulin clearance [31]. Elevation of hepatocellular lipid content, also called nonalcoholic fatty liver (NAFL), precedes the manifestation of T2D [12,32]. NAFL along with inflammatory processes in the liver often progresses to nonalcoholic steatohepatitis (NASH) and nonalcoholic fatty liver disease (NAFLD). All of these liver pathologies have been associated with mitochondrial abnormalities [33,34,35,36,37,38]. Nuclear magnetic resonance spectroscopy revealed a lower hepatic ATP content in patients with obesity-related NASH, and the capacity to recover from hepatic ATP depletion decreases along with an increase in body mass in patients and in healthy subjects [39]. Patients with metabolically well-controlled T2D showed reduced hepatic ATP concentrations that relate to hepatic insulin resistance independent of hepatic lipid content, suggesting that impaired hepatic energy metabolism and hepatic insulin resistance could precede the development of steatosis in patients with T2D [38]. A lower ATP production and lower ATP turnover have been found in the liver of diabetic patients with respect to non-diabetic individuals of comparable age and body mass [40]. A high-fat diet in young men caused liver IR, whereas peripheral tissues as mainly reflected by the muscles did not become IR [41], thus providing evidence to imply that increased hepatic lipid infiltration possibly occurs early in the pathogenesis [42]. High-resolution respirometry from liver biopsies showed higher maximal respiration rates in obese humans with or without NAFL compared with lean controls. These two groups were also compared with NASH patients, who showed higher mitochondrial mass, but lower maximal respiration, associated with hepatic IR and increased oxidative stress, suggesting an adaptation of the liver at early stages of obesity-related IR (hepatic mitochondrial flexibility), which is lost as the disease progresses [43]. A recent prospective study from the same group of researchers showed that during the first 5 years of T2D the increase in body fat leads to a doubling of liver fat content, whereas the energy metabolism of the patients’ livers progressively declines [44].

To decipher precise molecular mechanisms, cell models were exploited for investigation of direct effects of free fatty acids (FFAs) on hepatocytes. We and many others demonstrated that palmitate inhibits insulin-stimulated serine phosphorylation of Akt [45]; moreover, it has been shown that reactive oxygen species (ROS) played a causal role in palmitate-induced c-Jun NH2-terminal kinase (JNK) activation [46]. In addition, an inhibition of carnitine palmitoyl transferase-1 (CPT-1), which is the rate-limiting enzyme in β-oxidation, decreased palmitate-induced ROS production [46]. Furthermore, contrary to muscle, PGC-1α is upregulated in insulin resistant liver, and together with glucocorticoid receptor and hepatic nuclear factor-4a (HNF-4a), it activates the transcription of gluconeogenic enzymes and contributes to an increase of glucose output [47]. PGC-1α deficient mice showed improved hepatic insulin sensitivity [48]. Other important transcription regulators of genes for gluconeogenesis are members of the FoxO family of Forkhead transcription factors, which integrate insulin signaling with mitochondrial function in the liver [49]. Recently, two more factors from the same family have been described, FoxK1 and FoxK2, that are reciprocally regulated to FoxO1 following insulin stimulation and are involved in apoptosis, metabolism and mitochondrial function [50]. Mitochondrial-associated ER membranes (MAM), a cellular subcompartment important for lipid metabolism, is upregulated in hepatic IR [51,52]. Taken together, liver adiposity and mitochondrial function in hepatocytes play an important role in the development of insulin resistance and fatty liver disease in patients with diabetes.

### 2.3. Blood Cells

The main limiting factor for measuring mitochondrial respiration in humans is the invasiveness of tissue biopsy sampling. Thus, more recent studies started to investigate the use of isolated blood cells—peripheral blood mononuclear cells (PBMCs) and platelets—for ex vivo mitochondrial measurements in different diseases, including T2D. The first paper that aimed to measure mitochondrial parameters (membrane potential, mitochondrial mass and superoxide production) in PBMCs of type 2 diabetic patients was done more than a decade ago by Widlansky et al. [53]. They found smaller, more spherical mitochondria and decreased total mitochondrial mass in T2D patients, together with an increase of mitochondrial superoxide production and mitochondrial membrane potential in PBMCs of diabetic patients [53]. Two years later, another group assessed platelet mitochondria from diabetic patients and found lowered oxygen consumption, lower oxygen-dependent ATP synthesis, induction of mitochondrial anti-oxidant enzymes (superoxide dismutase 2 and thioredoxin-dependent peroxide reductase 3) and upregulation of oxidative stress, seen as increased protein carbonylation [54]. Basal, uncoupled and maximal oxygen consumption were higher in PBMCs from diabetic patients, which was consistent with higher ROS production [55]. In this study, the authors also suggested a link between mitochondrial and vascular smooth muscle cell dysfunction, due to a correlation between maximal oxygen consumption and non-endothelium-dependent nitroglycerin-mediated dilation [55]. A correlation study conducted in PBMCs from sedentary overweight/obese individuals, aged 68.3 ± 3.5 years, showed that higher maximal respiration and higher spare respiratory capacity of mitochondria from PBMCs was associated with better expanded short physical performance battery scores and lower plasma interleukin 6 [56]. The same group of researchers have continued with blood-based bioenergetic profiling (using PBMCs and platelets) and found it to correlate with brain glucose metabolism and frontal cortex mitochondria in non-human primates [56]. In a cohort of African Americans with longstanding T2D, some parameters of PBMC respiration showed positive correlation with brain morphology (white matter, grey matter and total intracranial volumes) [57]. Rose et al. [58] examined whether mitochondrial respiration in circulating cells (PBMCs and platelets) reflects that of skeletal muscle fibers derived from the same subject. They analyzed PBMCs, platelets and skeletal muscle samples in 32 women of varying body mass index with the use of extracellular flux analysis (Seahorse) and high-resolution respirometry (Oroboros) and found no correlation between respiration of circulatory cells and muscle, except for complex I LEAK and OXPHOS coupling efficiency, which correlated with permeabilized platelets and muscle [58]. However, to make definitive conclusion, both comparative studies and correlation analyses are needed to compare different measures of mitochondrial function from circulatory cells and biopsies and to correlate them with classical diabetes diagnostic parameters.

## 3. Interventions Targeting Mitochondria in Diabetes

### 3.1. Exercise

It has long been known that physical activity has many health benefits—it reduces risk and slows down progression of many chronic illnesses, such as diabetes, where it improves glycemic control [59] and insulin sensitivity in obesity [60]. Exercise exerts its effects mainly by acting on mitochondria; muscle tissue adaptation to exercise includes increased mitochondrial biogenesis and mitophagy [61]. Molecular mechanisms caused by muscle contraction that lead to an adaptive response in mitochondria are increased calcium concentration, increased AMP/ATP and NAD^+^/NADH ratios and increased ROS production [62,63]. These intracellular signals lead to activation of proteins responsible for regulation of mitochondrial biogenesis (PGC-1α) [64,65] and energy metabolism (AMPK) [66,67]. Transient, low-intensity ROS production caused by physical activity server as a signal that promotes beneficial effects, such as mitochondrial biogenesis and function (a phenomenon known as mitohormesis), contrary to the chronic elevation of ROS production caused by mitochondrial dysfunction and fuel overload that has a pathological effect contributing to insulin resistance and diabetes [68,69,70]. Studies show that moderate physical activity leads to an adaptive mitochondrial response not only in healthy individuals but also in pathological states where mitochondrial function is compromised such as aging, states that lead to muscle atrophy and chronic diseases in which there is mitochondrial dysfunction, so in this way exercise leads to a significant improvement in health [71,72,73].

The first evidence for the positive correlation between exercise and OXPHOS capacity, published in 1967, demonstrated that exercised rats exhibited a high level of respiratory control and tightly coupled respiration [74]. Holloszy et al. also showed that adaptations of skeletal muscle to endurance exercise is based on an increase in mitochondrial content and respiratory capacity of muscle fibers, which further affect substrate distribution: slower utilization of muscle glycogen and blood glucose, greater reliance on fat oxidation and less lactate production during exercise [75]. Exercise training was shown to increase in parallel mitochondrial function in vivo and insulin sensitivity measured by hyperinsulinemic-euglycemic clamp, but it did not change HbA1c or fasting glucose levels in T2D patients [71]. Using a combination of in vivo and ex vivo methods, after isolated isometric calf exercise, O_2_ availability was found to be the limiting factor for in vivo mitochondrial oxidative phosphorylation in sedentary adults with T2D—by supplementing O_2_ during and after exercise, mitochondrial function of diabetic subjects normalized and was not different compared with healthy adults of similar weight and activity level [29]. There are also findings that people with a single nucleotide polymorphism in the *NDUFB6* gene encoding a subunit of mitochondrial complex I do not respond to exercise by improving insulin sensitivity and glucose homeostasis [76], which is direct evidence that mitochondria are critical for exercise benefits in diabetes. Low-intensity exercises in healthy volunteers with a sedentary lifestyle increased PBMCs ROUTINE respiration, LEAK and OXPHOS with fatty acid substrates-dependent respiration by 31%, 65%, and 76%, respectively [77]. In addition, during 60 min of low-intensity exercise, a 2-fold higher lipolysis rate was observed, and 57% more fat was metabolized than during the incremental-load exercise [77]. High intensity interval training increased electron transfer (ET) capacity (respiration in the presence of uncoupler) in muscle of T2D patients and controls, irrespective of IR [78]. A recent study showed that, in a unique human in vivo model of unilateral lower-limb suspension with the contralateral leg serving as an active internal control, a low mitochondrial oxidative capacity due to physical inactivity directly impacts intramyocellular lipid accumulation, resulting in impaired insulin signaling upon lipid infusion [79]. This is a clear demonstration of the importance of decreased mitochondrial oxidative capacity and muscle fat accumulation in the development of insulin resistance in humans. Finally, in adults with T2D, single-leg exercise training augmented in vivo skeletal muscle oxidative flux and vascular content and function [80]. The authors commented that the improvement of mitochondrial respiration in skeletal muscle after this single-leg exercise training might be in part due to a better blood flow/oxygen delivery, and future studies will examine the time course of these two events [80].

Taken together what is written above and as recently reviewed [81,82,83], it is clear that exercise training improves metabolic health and insulin sensitivity in T2D patients. This occurs through different cellular mechanisms (Figure 1), and mitochondria are central for many of these processes.

### 3.2. Pharmacological Therapy

Numerous pharmacological agents used in therapy of diabetes and other diseases have recently been shown to impact mitochondria, and depending on the drug and specific study, this effect was either positive or negative, direct or indirect. Some drugs are specially designed to target mitochondria, and this might be a promising approach in the development of novel agents for diabetes therapy, as is discussed below. Many widely used antidiabetics have been reported to affect mitochondrial function; this is to be expected, as they usually target energy metabolism in some way. Having in mind the potential effects these and other drugs might have and how this could interfere with the results is especially important when designing studies that explore mitochondrial function and choosing study subjects, because it is not always possible or safe for the subjects to be taken off their standard therapy for the study, ideally with a prior wash-out period.

Metformin is an oral antidiabetic from the biguanide group; despite being widely prescribed for T2D treatment for over 60 years, its molecular mechanisms of action are still a matter of debate. Inhibition of respiratory chain complex I is one of the commonly proposed molecular mechanisms underlying the therapeutic effects of metformin observed in vivo—inhibition of gluconeogenesis in the liver and activation of glucose uptake in the muscle. It is proposed but not clear if metformin acts on insulin sensitivity in these tissues. Inhibition of mitochondrial respiration has been shown in some in vitro studies [84,85], but the metformin concentrations used were significantly higher than the plasma concentrations found in patients taking the drug. This led to this mechanism being questioned, as it has not been proven that lower (therapeutic) doses of metformin inhibit complex I, and some in vivo studies have shown that metformin does not alter normal skeletal muscle mitochondrial respiration in humans [86,87]. Larsen et al. [87] also showed that patients treated with sulphonylurea had the same respiratory rate as controls and patients in the metformin-treated group. Some more recent studies suggest a different mitochondrial mechanism of lower, therapeutic metformin concentrations—inhibition of mitochondrial glycerophophate dehydrogenase and consequential altered redox state was shown in primary rat hepatocytes, mouse hepatocytes and liver of metformin treated rats [88,89,90]. Wang et al. [91] showed that therapeutic metformin concentrations lead to increased mitochondrial respiration and ATP production in hepatocytes and increased mitochondrial density and complex I activity in liver of metformin-treated high-fat fed mice, all caused by an AMPK-dependent mechanism. Having this in mind, metformin is frequently mentioned as a respiratory complex I (CI) inhibitor and AMPK activator, especially in the context of exercise mimetics, a topic that is tackled later.

Thiazolidinediones (TZDs) or glitazones are antidiabetic drugs that act on peroxisomal proliferator-activated receptor (PPAR)-γ, primarily in adipose tissue, promoting the storage of fatty acids and inhibiting their export, thus being classified as PPAR agonists, but also as specific inhibitors of the mitochondrial pyruvate carrier (MPC) [92]. PPAR-γ regulates the transcription of many genes important for mitochondrial function and biogenesis, while mitochondrial pyruvate uptake is necessary for efficient gluconeogenesis, and thus for regulation of glycemia by the liver, and was shown to be aberrant in some diseases, including T2D [93]. However, potential effects of TZDs on mitochondrial function have been debated and explored. Results of these studies are not conclusive, and it seems that different TZDs could have opposite effects on mitochondrial respiration—one study showed that rosiglitazone decreased while pioglitazone increased mitochondrial respiratory capacity in skeletal muscle [94]. In contrast to this, pioglitazone was found not to alter ATP production in skeletal muscle of patients, despite increasing insulin sensitivity [95], while a different study showed that 6 months of pioglitazone treatment improved the mitochondrial proteomic profile, which is disturbed in T2D [96].

Sodium-glucose co-transporter-2 (SGLT-2) inhibitors or gliflozins, a recently approved group of antidiabetic drugs, have been shown to inhibit complex I respiration in primary mouse hepatocytes [97] and to increase expression of proteins important for mitochondrial biogenesis and function in white adipose tissue of mice with high fat diet—induced obesity and cultured adipocytes [98]. Moreover, it has been shown that the beneficial cardiovascular effect of SGLT-2 inhibitors in diabetic animal models is mediated through their effect on mitochondrial size and number as well as mitochondrial dynamics [99], but there are no data on their effect on mitochondrial respiration in humans. Due to the molecular mechanism of SGLT-2 inhibitors demonstrated in cell and animal models so far, gliflozins might be assigned a respiratory complex I inhibitor and/or PPAR agonist pharmacological action in the future.

Insulin is the major hormone responsible for regulation of glucose homeostasis, and it is known to increase glucose oxidation and storage. Despite this, the mechanistic effects of insulin on OXPHOS are not well understood, and data are lacking. Insulin is also used for treating diabetes, and the effects of long-term insulin therapy on mitochondrial function have also not been studied enough. It was shown that short-term insulin treatment acts on mitochondria by increasing coupling efficiency (decreasing LEAK respiration) in rat and human cultured myotubes [100]. There are some studies that show an increase of mitochondrial ATP production measured in vivo [101] and ex vivo in muscle biopsies of human subjects [102], but mainly after acute insulin infusion and not prolonged insulin treatment. Additionally, it would be difficult to discern if the potential effect on mitochondrial function was due to insulin treatment or improved glycemic control. However, insulin might be considered as an OXPHOS modulator in terms of its effect on mitochondria, but precise mechanisms are still to be defined.

Other medication used for treating diseases that are common comorbidities of diabetes, such as statins used for treating hypercholesterolemia, have been shown to have an effect on mitochondrial function as well [103]. Simvastatin inhibited maximal OXPHOS capacity with complex I and II-linked substrates, possibly as a consequence of reduced coenzyme Q10 content [104]. Another lipid-lowering drug is a nicotinic acid analog, a niacin derivative and NAD+ precursor acipimox, that has been studied for its effects on mitochondrial function. Because of its effects on lowering nonesterified fatty acids (NEFA), it was hypothesized that in diabetic patients it could have a beneficial effect on ectopic fat accumulation, mitochondrial function and IR. Despite the rebound effect that caused increased NEFA levels and decreased insulin sensitivity (the measurements were done one day after stopping the treatment), acipimox treatment caused an increase in OXPHOS capacity using complex I+II substrates, as well as maximal uncoupled respiration (ET capacity) with CI+II and fatty acid substrates, normalized to mtDNA copy number [105]. This study was the first to demonstrate a NAD^+^ booster effect of acipimox and its direct influence on skeletal muscle mitochondria in humans. On the other hand, a different study found that just one day of acipimox treatment improved insulin sensitivity but did not change mitochondrial respiration [106]. Interestingly, this study also examined insulin-stimulated mitochondrial function (4, 40, 100 nM/L ex vivo insulin stimulation before respiratory measurement on skeletal muscle biopsy) and showed that insulin stimulation had no effect on respiration in the T2D group and non-diabetic obese controls, while lean controls had an increase in respiration following insulin stimulation. Acipimox treatment did not change insulin-stimulated respiration either.

Table 1 summerises the most important findings of the above mentioned studies that explore pharmacological treatments, regarding their effects on mitochondrial function and insulin sensitivity (Table 1).

### 3.3. Novel Therapeutic Approaches That Target Mitochondria

Taking into account all studies mentioned above, mitochondria appear to be a valuable target for the development of novel drugs with a potential for a more integrative therapeutic approach, treating IR at different levels and in different tissues. The most promising pharmacological approaches to target mitochondria in different pathologies were recently reviewed [109]. Here we focus on mitochondrial targeting agents that might be relevant in therapy of diabetes and insulin resistance (Figure 2). Some of the potential pharmacological strategies shown in Figure 2 were mentioned in the previous Section 3.2 (CI inhibitors, AMPK activator, PPARs agonist, MPC inhibitors, OXPHOS modulator and NAD+ booster); some of them are described in this section as novel therapeutic approaches (antioxidants, sirtuin-activating compounds (STACs), mitochondrial permeability transition pore (mPTP) inhibitors), while a few of them are potential approaches not tested in diabetes yet, but are described as valuable targets for diabetes therapy (mitochondrial membrane properties modulators, ROS scavenger, CoQ10 (coenzyme Q10) analogues, mitochondrial-associated ER membranes (MAM) modulators), and there is still space for novel drugs to be designed. For some of the new substances, the mechanism of action has been proven only in animal models and in the few human studies that are reviewed here.

Certain substances, called exercise mimetics, mimic the effects of exercise by activating some of the same signaling pathways. Some of the most studied are AICAR, metformin (AMPK activators) and GW501516 (PPAR agonist), and some natural products found in certain foods, such as resveratrol (which can be found in grapes and red wine) and epicatechine (found in cocoa and dark chocolate). It has been shown that on the cellular level, exercise mimetics act by increasing mitochondrial content [110], activating fatty acid oxidation [111], reducing oxidative stress [112] etc. Exercise mimetics have a potential to act as a pharmacological supplement for exercise, especially considering that some people are not able to exercise due to illness.

Resveratrol was initially identified as a Sirt1-activating compound (STAC) [113]. Sirtuin (Sirt1) is a NAD^+^-dependent histone deacetylase important for promoting mitochondrial biogenesis and turnover. Given as an additional treatment with standard oral antidiabetics, resveratrol did not increase insulin sensitivity, but increased OXPHOS capacity with lipid-derived substrate (octanoyl-carnitine). Resveratrol supplementation did not change mitochondrial content (mtDNA copy number), PGC-1α expression, expression of respiratory chain proteins or PCr recovery time (in vivo respiration) [107]. Some studies also explored a combination of resveratrol and other polyphenols with antioxidant properties, such as epigallocatechin-3-gallate—this treatment increased some of the respiratory states (complex I+II OXPHOS capacity and ET capacity) in obese, non-diabetic subjects, without improving insulin sensitivity [108].

Imeglimin is a novel glucose-lowering drug that is currently being tested, with a few stage III clinical trials recently completed [114,115]. It was shown to be efficient in lowering fasting glucose and HbA1c in streptozotocin-treated rats and as monotherapy or in combination with other antidiabetics in human subjects [116], with few side effects [117,118,119]. Its dual effect, on both glucose-stimulated pancreas insulin secretion and insulin sensitivity in liver and muscle, makes it a great candidate for therapy, as it covers more than one pathological mechanism that leads to T2D. The proposed target of imeglimin action is the mitochondrion, and animal studies show that imeglimin treatment decreased respiration with complex I-linked substrates, while respiration with complex II-linked substrates was increased, in isolated mitochondria from liver of high-fat and high-sugar diet (HFHSD) mice. The same study shows an increase in complex III protein content and enzyme activity, hypothesizing that by inhibiting CI and restoring CIII function, CII-linked respiration is favored, which could result in increased fatty acid oxidation and decreased intrahepatic lipid accumulation. Imeglimin treatment also reduced ROS production at CII, which was increased in HFHSD mice, and as presumed by the authors, caused by reverse electron transfer from CII to CI (it could be eliminated by rotenone) [120]. Decreased ROS production by preventing reverse electron transfer was also shown using cultured endothelial cells, whose glucose-induced cell death was prevented by imeglimin treatment, by inhibiting formation of permeability transition pore but without any effect on mitochondrial respiration [121]. Imeglimin treatment was also shown to decrease glucose production and the ATP/ADP ratio and to increase mitochondrial redox potential in primary hepatocytes, without affecting mitochondrial respiration [122]. Due to all this, imegimin is a promising antidiabetic drug with potential OXPHOS modulating properties, possibly through CI inhibition, mPTP inhibition and antioxidant activity.

## 4. Conclusions and Perspectives

Mitochondria have been extensively studied in the context of T2D and have been recognized as an important factor in the development and progression of this disease, thus becoming a valuable target for lifestyle and pharmacological interventions. Discrepancies between the results of some of these studies might be due to their experimental design or different techniques used for assessing mitochondrial function, which makes it difficult to compare studies and draw a clear conclusion. Even when using the same methodology, researchers often use different terminology—for example, when describing respirometric data (reviewed in detail elsewhere [12,13])—which leads to a further problem in data reproducibility and comparability. As for study design, it is important to acknowledge that choosing study subjects and defining experimental groups is a crucial step that should be given a lot of thought. As diabetes is a very prevalent disease today, and as lifestyle and genetics play an important role in its pathogenesis, defining a healthy control group might be extremely complex. Obesity and low physical activity affect skeletal muscle function and energy metabolism in a profound way—when studying mitochondrial function in skeletal muscle in the context of diabetes, most studies take obese, non-diabetic (insulin sensitive) subjects for a control group or have more than one control group (for example, lean and obese controls). Taking in account genetic factors is even more difficult because genetic risk factors for diabetes are not so clear as the lifestyle ones—some studies compare people with diabetes, people who have first-degree relatives suffering from diabetes as well as healthy controls.

Here, we reviewed human studies investigating mitochondria in skeletal muscle and liver as well as in blood cells. In all these tissues, mitochondrial function is impacted in diabetic patients, but underlying tissue-specific mechanisms may differ. Related to the tissues’ role in glucose homeostasis, liver mitochondria are involved in hepatic lipid accumulation leading to an excessive fuel load to be processed [123]. Differently, skeletal muscle mitochondria are more affected during physical activity and are the most responsible for the health-promoting effects of exercise training, but ectopic fat deposition is also present in the muscle and is one of the mechanisms responsible for muscle IR. Interestingly, intramyocellular lipid accumulation was also observed in professional athletes, which is an adaptive mechanism to high energy expenditure, and in this case, it is not related to insulin resistance (athlete’s paradox) [124]. Mitochondrial dysfunction is also present in other tissues involved in diabetes pathogenesis and tissues affected by diabetes complications, but those issues were not included in this review (recently reviewed elsewhere) [14]. The role of mitochondria in different tissues in IR and diabetes is still to be investigated, and for this, animal and cell models could be of high relevance. These models are also crucial for exploring molecular mechanisms, drug development and preclinical studies for discovery of novel pharmaceuticals that target mitochondria.

As mentioned above, there is a problem of sampling human tissue for ex vivo analysis of mitochondrial function, mostly because of its invasive nature. For these reasons, blood cells are becoming more attractive for respirometric analysis as they are easily accessible and circulate through the whole organism, thus having a potential to reflect systemic metabolic changes. New clinical studies on bigger cohorts will demonstrate if there is diagnostic or prognostic potential of mitochondrial respiratory parameters of circulating cells and if they can be used as a surrogate marker for mitochondrial-targeted therapy.

As it is widely discussed, a balance between fuel availability and energy expenditure is necessary for optimal mitochondrial function, and any disturbance of this balance will challenge it. This can be compensated for to a certain point, but eventually mitochondria start to fail. The ability of mitochondria to cope with a metabolically changed environment is dependent on their respiratory capacity, which is variable and depends on lifestyle, age, genetic and epigenetic factors [9]. In T2D, there is an excess of fuel (glucose and fatty acids), and often a lack of energy expenditure (sedentary lifestyle), and this disbalance becomes a chronic state. The consequential mitochondrial dysfunction further propagates IR and ectopic lipid accumulation, creating a vicious circle. The good thing is that respiratory capacity can be improved by changes in lifestyle, the most important being physical activity and healthy diet, as well as pharmacological agents that target mitochondria (Figure 3). Many studies reviewed here demonstrate that by increasing mitochondrial respiratory capacity, certain interventions can cause an improvement of insulin sensitivity, glucose handling or ectopic fat deposition; thus, mitochondria might serve as a regulator of metabolic health and present a perfect target for novel therapies.

## Figures and Tables

**Figure 1 ijms-22-06642-f001:**
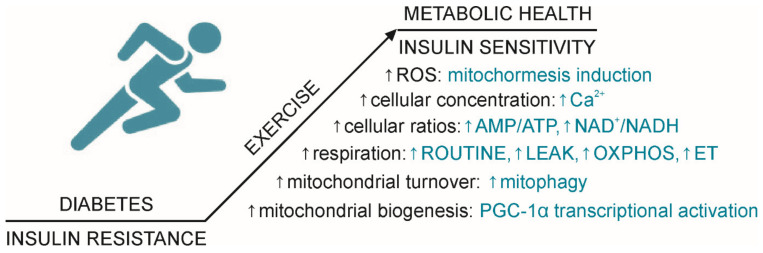
Cellular mechanisms underlying exercise as lifestyle intervention in T2D. Exercise increases insulin sensitivity in T2D by affecting energy metabolism through induction of mitochormesis, stimulation of mitochondrial turnover and biogenesis, increase in Ca^2+^ concentration, AMP/ATP and NAD^+^/NADH ratios and increase in respiration: ROUTINE, LEAK, OXPHOS and ET capacity. Symbol: ↑ = increase.

**Figure 2 ijms-22-06642-f002:**
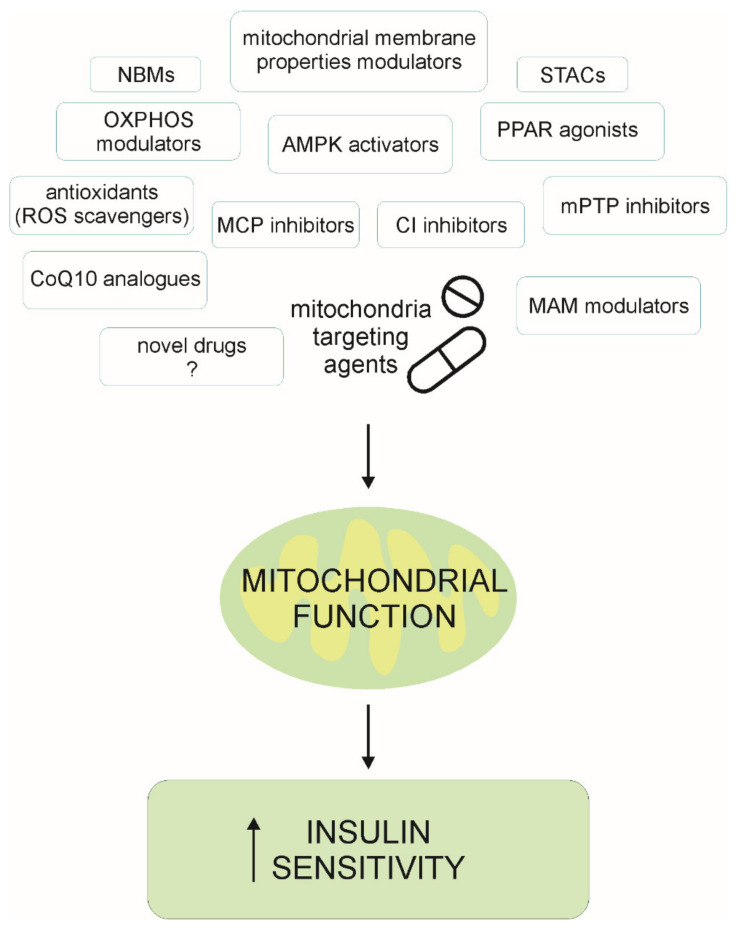
Potential mitochondria targeting agents that might improve insulin sensitivity through modulation of mitochondrial function. They are grouped based on the strategy of their pharmacological actions, which implies potential molecular mechanisms: NBMs (NAD+ boosting molecules), mitochondrial membrane properties modulators, STACs (Sirt1-activating compounds), OXPHOS modulators, AMPK activators, PPAR agonists, antioxidants (ROS scavenger), MCP (mitochondrial pyruvate carrier) inhibitors, CI (respiratory complex I) inhibitors, mPTP (mitochondrial permeability transition pore) inhibitors, CoQ10 (coenzyme Q10) analogues, mitochondrial-associated ER membranes (MAM) modulators and novel drugs to be designed. Symbols: ? = unknown drug—to be designed; ↑ = increase.

**Figure 3 ijms-22-06642-f003:**
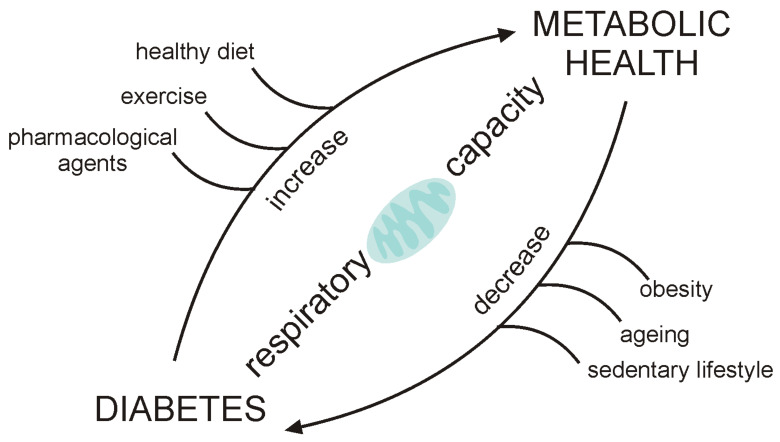
Mitochondria as an effector target of main diabetogenic factors such as obesity, aging and sedentary habits, which all cause a decrease of respiratory capacity, while the lifestyle and pharmacological interventions cause an increase of respiratory capacity and improvement of metabolic health.

**Table 1 ijms-22-06642-t001:** The effects of drugs on insulin sensitivity and mitochondrial function—a list of human studies with respirometric data. T2D, type 2 diabetes; OXPHOS, oxidative phosphorylation; ET, electron transfer; LEAK; CI, complex one of the respiratory chain; CII, complex two of the respiratory chain; M, malate; G, glutamate; S, succinate; Oct, octanoylcarnitine; EGCG, epigallocatechin-3-gallate; RES, resveratrol; ROSI, rosiglitazone; PIO, pioglitazone. Symbols: ↑ = increase; ↓ = decrease.

Substance	*n*	Treatment	Effects on Insulin Sensitivity	Effects on Mitochondrial Function	Reference
simvastatin	10 (hypercholesterolemia), 10 (healthy controls)	simvastatin (10 to 40 mg/day) for at least 12 months (average: 5 years)	↓ in patients compared with controls, not measured before and after treatment	↓ OXPHOS capacity (complex I and II-linked substrates) ↓ coenzyme Q10 content	Larsen 2013 [104]
acipimox	21 (T2D)	acipimox (250 mg three times daily) or placebo for 2 weeks	↓	↑ OXPHOS capacity (CI+II substrates) ↑ ET capacity (CI+II and fatty acid substrates)	Van der Weijer 2015 [105]
acipimox	15 (T2D obese)	acipimox (250 mg three times daily) for 1 day, and day after one dose (250 mg) 1 h before muscle biopsy or placebo	↑	= basal or insulin-stimulated mitochondrial respiration in permeabilized fibers ↓OXPHOS, ET capacity (octanoyl-carnitine), LEAK on acipimox in isolated mitochondria	Phielix 2014 [106]
resveratrol	17 (T2D)	resVida (150 mg/day trans-resveratrol) or placebo for 30 days	=	↑ OXPHOS (MOct, MGSOct) and ET capacity (M+O)	Timmers 2016 [107]
epigallocatechin-3-gallate and resveratrol	38 (overweight or obese, non-diabetic)	EGCG+RES (282 and 80 mg/d, respectively) or placebo for 12 weeks	=	↑ OXPHOS capacity (CI+II substrates and CI+II+fatty acids) ↑ ET capacity	Most 2016 [108]
metformin, sulfonylurea	22 (T2D), 18 (healthy controls)	T2D patients: metformin (2000 ± 200 mg/day, *n* = 14) or sulfonylurea (glimepiride, 2.4 ± 0.2 mg/day, *n* = 8)	not measured	=	Larsen 2012 [87]
metformin	18 (chronic heart failure, non-diabetic)	extended-release metformin (Glucophage XR^®^) (starting dose 500 mg q.d, uptitration to a target dose of 1000 mg b.i.d, *n* = 10) or placebo (*n* = 8)	=	=	Larsen 2021 [86]
pioglitazane	24 (T2D, obese)	pioglitazone (30 mg/day, increased to 45 mg/day if needed, *n* = 16) or placebo (*n* = 8) for 12 weeks	↑	= no change in muscle maximal ATP synthetic capacity (ATPmax) by 31P-MRS	Bajpeyi 2017 [95]
pioglitazone (PIO), rosiglitazone (ROSI)	21 (T2D), 8 (healthy controls)	T2D patients: ROSI (4 mg/d, *n* = 12), PIO (30 mg/d, *n* = 9), for 12 weeks	↑	↓ OXPHOS capacity (CI and CI+II) by ROSI treatment ↑ OXPHOS capacity (CI and CI+II)by PIO treatment ↑ protein content, complexe II and III by PIO treatment	Rabøl 2010 [94]

## Data Availability

Not applicable.

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
