# Peer review of "Targeting Mitochondria in Diabetes"

_ijms, 2021, doi:10.3390/ijms22126642_

Round 1
Reviewer 1 Report
A review by Jakovljevic et al. discusses the role of mitochondria in the development of diabetes mellitus. This review focuses on human studies. The authors consider the issues of mitochondria-targeted strategies for the treatment of diabetes mellitus.
Comments:
- Analysis of the literature on diabetes-induced mitochondrial dysfunction in different tissues is carried out quite often. Authors need to more carefully indicate the purpose of the paper. For example, the authors should indicate how this review differs from other similar works (PMID: 31652915; PMID: 32911736; PMID: 30601700), published in this journal as well.
- The list of references contains less than 40% literature sources (from 91 ones) for the period from 2016 to 2021. Based on this, the review does not look like a presentation of modern data.
- Line 230-233. “Molecular mechanisms caused by muscle contraction, that lead to an adaptive response in mitochondria are: increased calcium concentration, increased AMP/ATP and NAD+/NADH ratios and increased ROS production [41,42]”. The authors write that during physical activity there is an increase in ROS production. At the same time, it is known that oxidative stress is one of the main pathological factors in the development of diabetes mellitus. Will physical activity lead to more severe consequences at the cellular level in the development of T2D? The authors should describe in more detail how ROS production differs in physical activity and diabetes mellitus.
Author Response
Thank you for your kind review, please see the attachment with our point-by-point response.

Reviewer 2 Report
Summary: In this review article by Jakovljevic and colleagues human studies on the impact of insulin resistance and Type 2 diabetes on mitochondrial function in liver, muscle and blood cells as well as traditional and novel interventions (lifestyle or pharmacological) targeting mitochondria are discussed. The authors discuss studies showing that mitochondrial respiratory capacity decreases as the disease progresses, but increases following lifestyle (e.g. exercise) or pharmacological interventions. They suggest that novel pharmacological compounds that directly target mitochondria, could have great potential at treating both causative as well as secondary defects of diabetes. This review manuscript is very well written and it’s sections are well organized.
However, there are a few issues that should be addressed.
Major points.
Page 2: Lines 93-98: This section is poorly written and somewhat confusing.
Among all insulin-sensitive tissues, we will focus on the role of mitochondria in skeletal muscle as major site for glucose disposal and liver as major organ of endogenous glucose production, since insulin resistance and mitochondrial dysfunction in these two tissues have the biggest impact on whole-body energy homeostasis. Mitochondria in circulatory cells of diabetes patients have been studied in the recent years and we will here sum up literature findings in regard.
Page 4: Lines 172-175: The end of this sentence is unclear.
In this paper, authors reviewed some animal studies relevant to the conclusion that impaired energy metabolism in the liver could be an early defect in the pathogenesis of type 2 diabetes due to a negative correlation of hepatic ATP turnover with hepatic and whole-body IR respect to muscle IR.
Pages 5-6: Section 3.1. Exercise: This section on exercise could be improved by adding studies that correlate the beneficial effects of exercise on mitochondrial function with improvement in insulin sensitivity and glucose tolerance.
Here is one citation that could be added. The authors should also look for more recent studies.
(Menshikova et al. , Am J Physiol Endocrinol Metab (2005) 288(4): E818-25: Effects of weight loss and physical activity on skeletal muscle mitochondrial function in obesity)
Page 10: Lines 444-445: This sentence is not clearly written.
Other tissues involved in diabetes development and considered with their specific role on mitochondria have been recently reviewed [91].
Minor points.
Spelling mistakes
Page 2: Lines 61-62: …and when in the course of the disease does it arise - whether it precedes IR and hyperglycemia or does it come later.
Line 81: …in vivo measurement of ATP or phosphocreatine synthesis rates
Page 5: Lines 217-221: They analyzed PBMCs, platelets, and skeletal muscle samples in 32 women of varying body mass index, with the use of extracellular flux analysis (Seahorse) and high-resolution respirometry (Oroboros); and found no correlation between respiration of circulatory cells and muscle, except for complex I LEAK and OXPHOS coupling efficiency which correlated between permeabilized platelets and muscle [38]
Lines 243-245: …endurance exercise is based on an increase in mitochondrial content and respiratory capacity of muscle fibers, which further affect substrate distribution: slower utilization of muscle glycogen and blood glucose…
Page 6: Lines 284-285: …and depending on the drug and specific study this effect was either positive or negative, direct or indirect.
Lines 287-288: Many widely used antidiabetics have been reported to affect mitochondrial function;this is to…
Lines 296- 297: Inhibition of respiratory chain complex I is one of the commonly proposed mechanisms that explain its therapeutic effects – in addition to inhibition of gluconeogenesis in the liver and activation of glucose uptake in the muscle.
Page 7: Lines 310-311: Wang et al [63] showed that therapeutic metformin concentrations lead to increased mitochondrial respiration…
Page 10: Lines 421-422: …and defining experimental groups is a crucial step which should be given a lot of thought. As diabetes is a very prevalent disease today…
Author Response

(The authors gave the same response as above.)

Reviewer 3 Report
This review article addresses the mitochondria function in insulin resistance and type 2 diabetes. This review needs detailed explanation in the potential mechanisms linking mitochondrial function and insulin resistance or type 2 diabetes in different tissues such as muscle, liver and blood cells. Moreover, authors need to search more references to support the scientific understanding.
Comments are suggested as follows;
-This article does not provide the clear and comprehensive information on mitochondrial function in the context of diabetes. Therefore, authors should present the specific potential molecular mechanisms explaining the role of mitochondria in skeletal muscle, liver and blood cells in the context of diabetes by searching more references.
- Specific and clear molecular explanation mentioned in the text is required in Figure 1.
-Table 1 needs a legend including abbreviations.
-Please use symbols (e.g., ↑, increased) rather than words.
-In line 370, several references are needed for animal models and human studies.
-Authors are required to present the diagram which explain the potential mechanisms of the effect of drugs on insulin sensitivity and mitochondrial function.
Author Response

(The authors gave the same response as above.)
